# How the Lagged and Accumulated Effects of Stress, Coping, and Tasks Affect Mood and Fatigue during Nurses’ Shifts

**DOI:** 10.3390/ijerph17197277

**Published:** 2020-10-05

**Authors:** Fermín Martínez-Zaragoza, Jordi Fernández-Castro, Gemma Benavides-Gil, Rosa García-Sierra

**Affiliations:** 1Department of Behavioural Sciences and Health, University Miguel Hernández, 03202 Elch, Spain; f.martinez@umh.es (F.M.-Z.); gbenavides@umh.es (G.B.-G.); 2Departament de Psicologia Bàsica, Evolutiva i de l’Educació, Universitat Autònoma de Barcelona, 08193 Barcelona, Spain; 3Research Support Unit Metropolitana Nord, University Institute Foundation for Research in Primary Health Care Jordi Gol i Gurina (IDIAPJGol), 08303 Mataró, Spain; yedra4@hotmail.com; 4Department d’Infirmeria, Universitat Autònoma de Barcelona, Campus de Bellaterra, 08193 Barcelona, Spain

**Keywords:** stress, nurses, coping, mood, fatigue, burnout, ecological momentary assessment, lagged effects, accumulated effects

## Abstract

Nurses experience significant stress and emotional exhaustion, leading to burnout and fatigue. This study assessed how the nurses’ mood and fatigue evolves during their shifts, and the temporal factors that influence these phenomena. Performing a two-level design with repeated measures with moments nested into a person level, a random sample of 96 nurses was recruited. The ecological momentary assessment of demand, control, effort, reward, coping, and nursing tasks were measured in order to predict mood and fatigue, studying their current, lagged, and accumulated effects. The results show that: (1) Mood appeared to be explained by effort, by the negative lagged effect of reward, and by the accumulated effort, each following a quadratic trend, and it was influenced by previously executing a direct care task. By contrast, fatigue was explained by the current and lagged effect of effort, by the lagged effect of reward, and by the accumulated effort, again following quadratic trends. (2) Mood was also explained by problem-focused and emotion-focused coping strategies, indicative of negative mood, and by support-seeking and refusal coping strategies. (3) Fatigue was also associated with direct care and the prior effect of documentation and communication tasks. We can conclude that mood and fatigue do not depend on a single factor, such as workload, but rather on the evolution and distribution of the nursing tasks, as well as on the stress during a shift and how it is handled. The evening and night shifts seem to provoke more fatigue than the other work shifts when approaching the last third of the shift. These data show the need to plan the tasks within a shift to avoid unfinished or delayed care during the shift, and to minimize accumulated negative effects.

## 1. Introduction

Nurses are exposed to significant stress at work, which can lead to burnout, provoking a deterioration in nurses’ health, a decrease in the quality of their care, and an increase in their intention to quit [1,2,3,4,5]. Two widely studied models of work stress have been tested in the context of nursing—Karasek’s Demand-Control model [6] and Siegrist’s Effort-Reward Imbalance (ERI) model [7]. Thus, nurses suffering high demands or an excessive workload and with little control or autonomy when carrying out their tasks tend to present greater health problems [8,9,10]. In addition, the lack of balance between the effort made at work and the rewards obtained (i.e., recognition, salary, and opportunities for promotion) would also generate a greater risk of burnout in the medium and long-term, and greater vulnerability to different pathologies [11,12,13,14]. By combining both of the aforementioned models in tasks and situations that involve high levels of demand/effort with low levels of control/reward, negative mood increases significantly. However, if the level of control/reward was high, then the mood remained positive, even if the degree of demand/effort was high [15].

Fatigue and negative mood are considered to be an effect of work stress. Some of the consequences of fatigue are lapses in attention, an inability to stay focused, increased reaction time, reduced processing capacity, confusion, compromised problem solving, memory lapses, impaired communication, irritability, indifference, and lack of empathy [16,17]. Moreover, fatigue has been related to excess effort and lack of reward [18]. In the nursing setting, some interesting results link fatigue and mood to task demand, control, effort and reward. For example, high demands and lack of control or autonomy have an absorbing effect on fatigue [19]; nursing tasks are less rewarding and mood is worse when exhaustion or fatigue are detected [20]; and mood decreases with tasks that involve high levels of demand/effort and low levels of control/reward, whereas mood remains positive when the level of control/reward is high, even when the degree of demand/effort is also high [15].

It is also important to take into account how the fatigue and mood levels in nurses change throughout the work shift and the consequences of this. A decrease in the effectiveness of clinical decision-making as the working day progresses and over multiple work shifts has been observed [21,22]. The effort associated with long and continuous work shifts provokes high levels of fatigue, which negatively influence performance and patient safety [23,24,25]. Variations in mood have also been observed between the beginning and end of each shift [26,27]. Significantly, attempts have been made to predict the levels of fatigue from the values of physical energy expended, as well as the demand, control, and reward accumulated in the previous 90 min, through ecological momentary assessment throughout a nurse’s work shift. Results show that fatigue does not necessarily depend on the physical energy expended or on the perceived demands but, rather, on the perceived control over the work and the rewards associated with it [28].

However, nurses must perform different tasks in their work and not all of them have the same implications or are perceived in the same way. In this regard, physically demanding patient care tasks, logistic/management/organization tasks, and multitasking demands are those that appear to contribute to fatigue most frequently [29,30].

Likewise, not all coping strategies work in the same way in all situations and/or for all people, such that the risk of physical and psychological pathologies increases with the use of maladaptive coping strategies, including burnout [31,32,33,34,35,36]. Whilst problem-focused coping behaviours are not associated with fatigue in nurses, avoidance coping such as drinking alcohol, avoiding situations, or blocking emotions appear to predict fatigue [37]. Moreover, the use of avoidance coping and the perception of excessive demands on nurses are thought to be important predictors of mood disturbances [38]. Indeed, an important challenge in stress research is to understand how stress, coping, and health change in individuals over time [39], with insufficient progress in this field to date.

Therefore, it is not yet established to what extent the mood and fatigue that nurses experience during their work shifts are determined by the delayed and accumulated effect of the type of task performed; the perception of demand, control, effort, and reward of the task; and the type of coping used when performing them. Analyzing the influences between these variables in real time by ecological momentary assessment, is necessary to be able help improve the performance of nursing staff and their physical and emotional well-being, this being the main objective of the present study.

## 1.1. Aim/Research Questions

The initial aim of this study was to examine the relationship between stress, mood, and fatigue in nurses. A recent study concluded that fatigue does not depend on the demands or on the energy expended but, rather, on the perceived control and its associated reward [28]. Thus, it remains unclear if previous (lagged effect) or accumulated stress (demand/control, effort/reward) influences the mood and fatigue of nurses. Thus, we hypothesized that:
Accumulated effort will worsen mood and will increase fatigue, while reward will produce the opposite effect, as predicted by Siegrist’s model.

The second aim was to study the effect of coping on mood and fatigue. It has been shown that higher mean scores in problem-solving, avoidance, and seeking social support were associated with higher levels of compassion fatigue [40]. Thus, the question remains as to whether the coping strategy previously used or accumulated over time affects the mood and fatigue of nurses. Hence, we hypothesized that:
2.The use of problem-focused and seeking-support strategies will have a delayed and accumulated effect on mood and fatigue, producing an improvement in both; and3.The use of emotion-focused and refusal strategies will have a delayed and accumulated effect on mood and fatigue, worsening them.

The third aim of this study was to examine the effect of the task on mood and fatigue. It has been shown that physically demanding patient care tasks most frequently produced physical and mental fatigue in nurses [29]. Thus, the question arises as to whether the previous task or those accumulated over time affect the mood and fatigue of nurses. Accordingly, we hypothesized that:
4.Accumulated direct care tasks improve mood and increase fatigue;5.Administrative and organizational tasks will have a delayed and negative effect on mood, presumably due to tedium.

## 2. Materials and Methods

### 2.1. Design

Within-person and between-person relationships were identified by multilevel statistical analyses. A two-level design with repeated measures was used [41], in which level 1 was established from the moments at which the outcome variables were measured. In level 2, the moments were nested at the person level.

## 2.2. Participants, Settings, and Procedure

A random sample of 113 nurses was recruited from the following wards at two University hospitals, selecting 80% of the nurses on each ward: internal medicine, surgery, traumatology, oncology, cardiology, neurology, nephrology, pneumology, rheumatology, digestive, gynaecology, geriatrics, palliative care, paediatrics, and psychiatry. Critical care services and emergency services were excluded from the cohort due to the distinctive tasks involved. The nurse–patient ratio ranged from 10 patients in the day shift to 30 patients in the night shift. Of the nurses recruited, 17 finally refused to participate in the study, and thus the final sample was 96 nurses with a response rate of 85% (96 of possible 113). Data were collected between January and December of 2015, excluding holiday periods.

## 2.3. Measures

### 2.3.1. Level 1 Measures (moment)

#### Ecological Momentary Assessment

Stone and Shiffman [42,43] have described ecological momentary assessment, in which data of interest are recorded frequently or intensively, in real time, and in a critical environment.

A Samsung Galaxy Mini Smartphone with an Android operating system was specially developed for this study and used to obtain ecological momentary measurements (all the measures in this study). The nurse carried the device during five consecutive work shifts. Data entry was prompted by vibration or a buzzing alarm and, if busy, the nurse could postpone the response for 10 min. This meant that in terms of hygiene, if the task they were involved in was direct care, it could be completed and the nurses could wash their hands before touching the screen. However, if the question remained unanswered for 20 min then this moment was registered as missing data. Answers were presented on analogue scales and the participants were given “tips” as to how to select their responses to the questions. The software was designed with a menu to help answer any queries by just touching the screen. The measurements taken at each evaluation point are listed below.

**Demand, effort, control, reward.** Four questions were designed to appraise the different aspects of work stress: demand, control (labelled as autonomy and skills development), effort, and reward. Each question labelled the term to be evaluated and was followed by a simple question as to how far each concept could be applied to the characteristics of the task being performed at the time. The four questions were answered using a visual analogue scale from 0 to 10 in order to evaluate the intensity of the response, one of the most usual response formats for single-item questions [44].

**Momentary coping.** A 10-item coping questionnaire was designed ad-hoc to assess nursing coping in an ecological momentary assessment context (*MoCoping*) [45] and based on the COPE Inventory [46,47]. The questionnaire followed a structure of one item for each strategy, grouped into four types: a problem-focused approach (including one item of active coping and one of planning), an emotion-focused approach (including one item of acceptance, one of reinterpretation, and one of distraction), support-seeking (including one item of emotional support and one of instrumental support), and a refusal approach (including one item of denial, one of venting, and one of self-blame). The variables were coded as binary—i.e., the coping strategy was used or not.

**Nursing task.** The task the nurse was involved in was coded based on an adaptation [20] of the WOMBAT classification (Work Observation Method by Activity Timing: [48,49,50]), and classified as: direct care, indirect care, medication, documentation, communication, and social/resting tasks. WOMBAT establishes 10 categories: direct care (communication with patient or family, applying dressings, bathing, etc...), indirect care (planning care, reviewing results, etc...), medication tasks (medication preparation, administration, documentation, etc...), documentation (on paper or electronic), professional communication (with other healthcare professionals), ward-related activities (coordinating beds, coordinating staffing, etc...), in transit (time between patients and between tasks), supervision (of students, nurses newly introduced to the service, etc...), social (breaks, meal times, etc...), and other. The variables were coded as binary—i.e., they are undertaking the task or not.

**Mood.** Mood was measured on a single-item, five-point visual analogue scale from a happy face to a sad face, where higher values reflect a poor mood (see Figure 1).

**Fatigue.** Fatigue was measured by a single-item, five-point visual analogue scale from a full battery to an empty battery, where higher values mean more intense fatigue (see Figure 1).

The order of each record at the person level (level 1, moment) was automatically recorded by the device, registering a mean of five records per shift (every half an hour).

### 2.3.2. Level 2 Measures (Person)

#### Questionnaires

**Ad-hoc questionnaire.** We recorded the work shift, gender, age, marital status, number of children, years of experience, and professional status of the subjects. The ad-hoc questionnaire was administered just before the smartphone was programmed in a room prepared for it.

### 2.4. Data Analyses

No missing data imputation was carried out and descriptive statistics were obtained using the R Statistical Package [51].

#### 2.4.1. Multilevel Modelling (MLM)

Multilevel analysis allows to control for the variance associated with random factors without data aggregation. As fixed effects, we entered the moment of time, demand-control, and effort-reward, and four different types of coping strategies for momentary use, nursing tasks, mood, and fatigue into the model. As random effects, we tested random intercepts and slopes for the effect of the same variables, allowing these to be varied randomly across groups. *p*-values were obtained by likelihood ratio tests of the full model with the effect in question compared against the model without the effect in question. Z-values were obtained to test the significance for fixed effects (estimates and standard error data in the tables). Trends in the longitudinal data were studied by testing the linearity of the model for all the outcomes.

#### 2.4.2. Assumptions

Visual inspection of q-q plots using the car R package [52] did not reveal any obvious deviations from homoscedasticity or normality in the dependent variables. The momentary coping strategies were categorized and coded as dummy variables (logit models), with mood and fatigue as quantitative variables.

#### 2.4.3. Initial Models, Model Fit, and Fit Criteria

Model comparison was achieved following the guidelines set out by Bliese and Ployhart [53] and Bliese [54], beginning the process by examining the nature of the outcome. To test the significance of the person effects, we carried out a likelihood ratio test comparing the null multilevel model (unconditional model) with a null single-level model, thereby testing the null hypothesis that there were no group differences. Subsequently, the intraclass correlation coefficient (ICC) was estimated in order to calculate the between/within variation ratio. The model’s fit was assessed using chi-squared tests on the log-likelihood values to compare the different models, and using the Akaike’s information criterion (AIC: [55]), a relative goodness of fit index. According to the change in these fitness indices, the last model with significant changes was chosen for each analysis.

The data were analysed using the R Statistical Package [51]. The lme4 R package [56] was used to analyse the quantitative outcomes variables and to obtain a multilevel longitudinal growth curve model of the relationship between stress (demand/control, effort/reward), coping strategies (momentary), tasks, and mood and fatigue. Linearity trends were also analysed by studying the quadratic and cubic trends, and interactions between time and predictors were examined. The *p*-values of the lme4 outputs were obtained with the lmerTest package [57], the ICC was calculated using the sjstats package [58], and graphic data processing was performed with the ggplot2 [59] R package.

#### 2.4.4. Mood and Fatigue from Lagged and Current Values of the Predictors

To predict the lagged effects on mood and fatigue, the lag-1 measures of the predictors were calculated (i.e., the measures taken in the moment immediately prior to the outcome measure). The current and lagged effects of the predictors were included in a two-level model with the observations over time nested into participants. These analyses were divided into three parts depending on the type of predictors used: stress, coping, or task. Each analysis was performed separately with each dependent variable, such that six analyses were performed to study the lagged effects. The predictors included were: stress (first analysis) (intercept; time; and current and lag-1 measures of demand, control, effort, and reward), coping (second analysis) (intercept; time; and current and lag-1 measures of problem-focused, emotion-focused, support-seeking, and avoidance), task (third analysis) (intercept; time; and current and lag-1 measures of direct care, indirect care, medication, documentation, social, and communication). Intercept and time were allowed to vary randomly, while the rest of the variables were considered as fixed effects.

#### 2.4.5. Mood and Fatigue from Accumulated Values of the Predictors

The accumulated values were calculated by summing the previous values of the variable at each time point within each shift. The analyses were again divided into three parts, depending on the type of predictors used: stress, coping, and task. Each analysis was performed separately with each dependent variable, such that six analyses were used to study the accumulated effects. The effects of the predictors were included in a two-level model with the time of observation nested into the participants. The predictors included were: stress (first analysis) (intercept; time; and accumulated demand, control, effort, and reward), coping (second analysis) (intercept; time; and accumulated problem-focused, emotion-focused, support-seeking, and refusal coping), task (third analysis) (intercept; time; and accumulated direct-care, indirect care, medication, documentation, social, and communication). Intercept and time were allowed to vary randomly, while the remaining variables were considered as fixed effects.

### 2.5. Ethics

All the nurses participating in the study provided their informed consent, and they were aware that they could leave the study at any moment. Approval was granted by the Ethics and Clinical Research Committee (CEIC) at the University Hospital of Elche (Spain) (CEIC-HGUE 23/01/2013) and at the Hospital of Terrassa (Spain) (CEIC-CST 18/06/2013).

## 3. Results

### 3.1. Sample Description

In the cohort of nurses studied, 89.90% were female, with a mean age of 40.22 years (*SD* = 8.50). In terms of the shift patterns, 45.45% worked rotating shifts and the remainder worked fixed shifts—either mornings, evenings, or nights. Moreover, 5.20% of the nurses had an additional job. The distribution between the two hospitals was 47.42%/52.58%, and the mean number of years the nurses had been working at their current job was 9.86 (*SD* = 7.99), with a mean of 17.40 years of experience as a nurse (*SD* = 8.36). Tenured staff made up 77.31% of the sample and 52.53% had a specialty in nursing beyond their university degree. In addition, 50.51% were in a relationship; 16.16% were single; and the remaining were separated, divorced, or widowed. In terms of the coping strategies used, problem-focused coping was used in 46.23% of the moments measured, emotion-focused in 44.89%, support-seeking in 8.13%, and avoidance in 2.70%.

### 3.2. Data Analyses

Initially, the effects of the work shift on the dependent variables were described prior to testing the hypotheses, and the data were analysed to see if there were any interaction effects between work shift and the time into the shift when fatigue can be predicted (see Johnston et al. [28]). Mood was explained by the effect of time (*E* = 0.09, *p* < 0.01), and its interaction with work shift was only marginal (*E* = 0.07, *p* = 0.07 for the evening work shift, and *E* = 0.07, *p* = 0.09 for the night work shift), displaying a linear trend with an ICC = 0.37. Fatigue was explained by the effect of time (*E* = 2.36 × 10^−1^, *p* < 0.000) and its interaction with the work shift (*E* = 9.83 × 10^−2^, *p* < 0.01 for the evening work shift, and *E* = 1.30 × 10^−1^, *p* < 0.01 for the night work shift), again with a linear trend and an ICC = 0.42.

The evolution of mood and fatigue during the work shift was assessed and negative mood increased gradually as the work shift progressed, with slightly higher values for evening and night shifts (Figure 2). In the case of fatigue, the values were again higher for evening and night work shifts, yet with the peculiarity that they increased from moment 4 to 5 (the end of the shift), although this remained stable at these two moments of the shifts in the morning and rotatory shifts.

The data from the final models of all 12 of the fitted models was organized in terms of the dependent variable and type of predictor (Table 1). The ICC of the fitted models is close to 0.50, indicating that the moment (level 1) and person (level 2) exert a similar influence. Significant effects of time were found in all the models, and there were significant fixed effects of time in all cases. Hence, negative mood and fatigue increased with time into the shift for all the predictors studied. The effects of all the variables depended on the person (see models of random effects).

**Aim 1:** Does previous or accumulated stress (demand/control, effort/reward) influence the mood and fatigue of nurses?

Mood can be explained by the current effort and by the negative lagged effect of reward, following a convex, quadratic trend (Table 1, first section, and Figure 3, first spaghetti plot), indicating that mood is more positive when there was a previous reward. The reward experienced previously was effective in reducing negative mood up to a certain extent, after which it remained stable. When stress accumulated, mood was explained by the accumulated effort according to a concave, quadratic trend (Table 1, first section and Figure 3, second spaghetti plot). Hence, when effort grows, negative mood was only enhanced up to a specific value, after which it started to decrease mildly.

Fatigue could be explained by the current and lagged effect of effort, and by the lagged effect of reward. When stress accumulated, fatigue was explained by the accumulated effort following a concave, quadratic trend (Figure 3, third spaghetti plot), while accumulated reward diminished fatigue (Table 1, first section). Accumulated effort also worsened mood only up to a given value, after which it decreased mildly.

**Aim 2:** Does the coping strategy previously used or accumulated over time influence the mood and/or fatigue of nurses?

Mood was explained by problem-focused and emotion-focused coping, and the use of these strategies indicated negative mood. When the effect was accumulated, mood was explained by emotion-focused and refusal coping strategies (Table 1, second section). Fatigue was explained by previous problem-focused coping, and by emotion-focused and refusal coping. When the effect was accumulated, fatigue was predicted by the four coping strategies assessed: problem-focused, emotion-focused, seeking support (negative), and refusal (Table 1, second section). Hence, the only coping strategy that diminished fatigue was the accumulated use of seeking support.

**Aim 3:** Does the task previously carried out or accumulated over time influence the mood and fatigue of nurses?

Mood was explained by the effect of executing a previous direct care task, and in terms of an accumulated effect only the documentation task explains mood, albeit negatively (Table 1, third section). Hence, being occupied with a documentation task for some time improved mood. Fatigue was explained by the effect of the direct care task, and by the prior effect of documentation and communication tasks—both in a negative sense, whereby the person involved previously in these tasks experiences less fatigue in the following moment measured. When the effect was accumulated, fatigue was explained by direct care and medication tasks (Table 1, third section).

## 4. Discussion

This study set out to examine the variables related to fatigue and mood in nurses in terms of the effects of their work shifts. The effects of the variables were analysed as each shift evolves, and this time analysis was especially relevant in the work of nurses, which is mostly carried out in response to demand. The role that these variables play in the long-term development of burnout is also important. In the first place, this study confirms that the effect of the work shift on fatigue depends on the time into the shift. As such, the evening and night shifts as opposed to the morning and rotatory shifts, seem to provoke more fatigue than the other work shifts when approaching the last third of the shift, results that are in line with earlier findings [28].

### 4.1. The Effect of Stress on Mood and Fatigue

While the first hypothesis was confirmed, the relationship detected was not lineal and the effect of stress on mood was, to a large extent, predicted by the Siegrist model [7]. When current effort is assessed when no previous reward was evident, mood is more negative. However, only accumulated effort produces negative mood. These relationships are not lineal and their effect only persists to the point where no more negative mood is produced despite increasing effort. This result fits with the nonlinear dynamics of effect that is well known in experimental studies in the field of neuroscience [60] and that is explained by emotional regulation mechanisms; it will be necessary to obtain more evidence in realistic studies to confirm this effect. The effect of stress on fatigue was predicted with precision by the Siegrist model. When prior or current effort increases and reward diminishes, fatigue increases. If effort and reward accumulate, the behaviour of these variables persists up to a given point, at which more effort or less reward does not produce more fatigue. Therefore, the effort during the first few hours of the shift serve as a “warm-up”, making it important to reduce fatigue later. These results are congruent with those found by Johnston et al. [28] about the role of reward in the perception of fatigue but not with demand or effort, which in the present study appear to be as important as reward.

### 4.2. Effect of Coping on Mood and Fatigue

The second hypothesis was not confirmed, as only the effect of seeking support reduced fatigue. The first impression regarding coping is that the data do not agree with the extensive literature indicating that active strategies are more adaptive than passive ones [61,62]. This discrepancy is undoubtedly due to how this was assessed in this study. The general style trend of coping was not recorded, but rather how each situation was coped with at the time it occurred, and the nurses could answer that they had not coped in any specific way. As such, it is not strange that mood and fatigue are worse after stressful situations that were coped with in a specific manner. Thus, this study offers evidence of a delayed effect of these coping styles. If we examine mood, we see that coping based on the problem has a delayed effect, whereby if you have had to cope with a problem it continues to have an influence for some time later, even though repeatedly facing a problem does not have a negative effect. Alternatively, the search for social support has no positive or negative effect on the subsequent mood.

In terms of fatigue, we see that situations that need to be addressed increase fatigue, although the search for social support, when repeated, reduces fatigue. The buffering effect of social support in fatigue has been consistently tested in chronic patients [63,64]. Thus, an important means to control fatigue is to broaden and improve the social support networks between professionals on the ward and in the hospital. Social support is not only offered by co-workers [65], yet the support from the nurse’s manager was significantly associated with their self-perception of fatigue. In this sense, the American Organization for Nursing Leadership recommends to allocate resources to support nursing, as one of the necessary elements for a Healthy Practice Environment [66]. Integrating social support into daily practice in a sustainable way involves a process of routinizing and sustaining the new practice within each particular social context [67].

In general terms, the third hypothesis is confirmed, since the use of emotion-focused and refusal strategies worsens the mood and increases fatigue. However, while refusal had cumulative effects it had no delayed effects. Hence, rejecting a stressful situation is not related to being in a worse situation a little time later, after it has been avoided, yet rather its repetition enhances negative mood. Therefore, problem-based coping has an opposite effect to that of the refusal, and while in the short term it can decrease well-being and it may be implemented without negative consequences, refusal instigates short-term well-being but it is not a resort used repeatedly. Refusal coping has an important and negative impact on the generation of stress and depression, as it is known [68].

### 4.3. Effect of Nursing Task on their Mood and Fatigue

The fourth hypothesis was confirmed only in part, as direct care tasks worsen mood (in contrast to the expected effect) and they produce more fatigue. When engaged previously in a direct care task, the nurse’s mood becomes more negative, even though direct care tasks are generally reported by most nurses as very core-professional tasks. It might also have been of interest to know if the direct care task undertaken had or had not been finished by the nurse at the time of reporting [69]. It may be concluded that the pressure of care and tasks that require immediacy is detrimental to the direct care tasks, to which the nurse may dedicate less time than they consider necessary. The explanation for this result can be found in the pressures on time usage, which could precipitate implicit rationing by establishing clinical priorities in the nursing setting, resulting in staff feeling that some care tasks may have been left undone, a situation referred to as “unfinished nursing care” [70]. For nurses, satisfaction in daily task accomplishment is negatively associated with changes in negative affect at the end of the shift. Moreover, the impact of task accomplishment satisfaction on changes in daily affect is larger for direct care nursing tasks than for the rest of nursing tasks [69]. Thus, having the feeling at the end of the shift of unfinished nursing care when the majority of the tasks have been related to direct care would be the circumstance producing the strongest negative affect. However, direct care tasks have a negative delayed effect on mood but not a cumulative effect. This may reflect that it is worse to alternate direct care tasks with other types of tasks than to be able to focus on the demands of direct care. As for mood, direct care produces delayed effects on fatigue. However, both direct care and medication have affected fatigue, which does not occur with mood. Therefore, concentrating on a task when alternating with others can help improve mood at the cost of increasing fatigue. This effect is consistent with experimental studies on sustained attention when performing tasks [71] that explain how sustained attention hinders performance and distraction is a relief.

The fifth hypothesis was not confirmed, as administrative and organizational tasks such as documentation or communication seem to improve fatigue, and when documentation tasks accumulate, mood is more positive. Documentation or communication are apparently unpleasant tasks for nurses, and they are considered to be best undertaken when they are not very tired. Since there are no studies that analyse these subtle differences between nursing tasks, it is only possible to offer this speculative explanation that, of, course should be confirmed in future research. Communication with other healthcare professionals is often considered a complicated task, yet it appears to be a more relaxing task that produces positive mood. Some studies have found that 20% of the shift time of nurses is dedicated to the communication task [50], so it will important in the future to think over the role and importance of this task.

### 4.4. Limitations and Perspective for Future Studies

Perhaps the main limitation of this study is that the results and conclusions are only generalizable to ward nurses working in hospitals. Other tasks performed by different types of nurses may differ in terms of the effects of the different coping strategies used. Nevertheless, the vast majority of registered nurses who work in hospitals are ward nurses, so this study covers their professional profile quite well. Future studies should also focus on mechanisms of recovery from fatigue and from negative mood, such as resting and sleep.

## 5. Conclusions

These results of this study indicate that mood and fatigue do not depend on a single factor, such as workload, but rather on the distribution and evolution of tasks throughout the shift, the stress they cause, and how it is coped with. These data show the need to plan the tasks within a shift to avoid unfinished or delayed care during the shift, and to minimize accumulated negative effects. In addition, the introduction of facilities for routine social support, both by co-workers and supervisors, is also important in order to prevent burnout.

## Figures and Tables

**Figure 1 ijerph-17-07277-f001:**
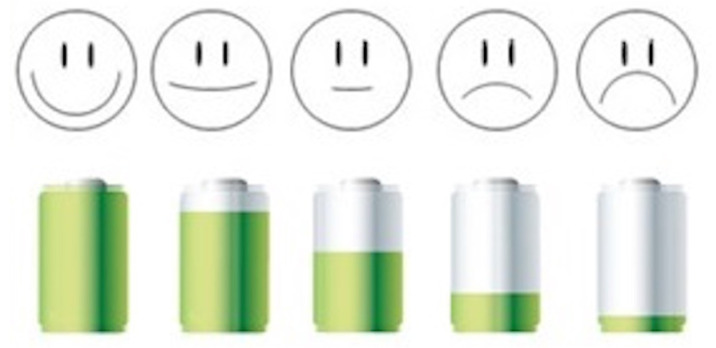
Visual analogue scales for mood and fatigue.

**Figure 2 ijerph-17-07277-f002:**
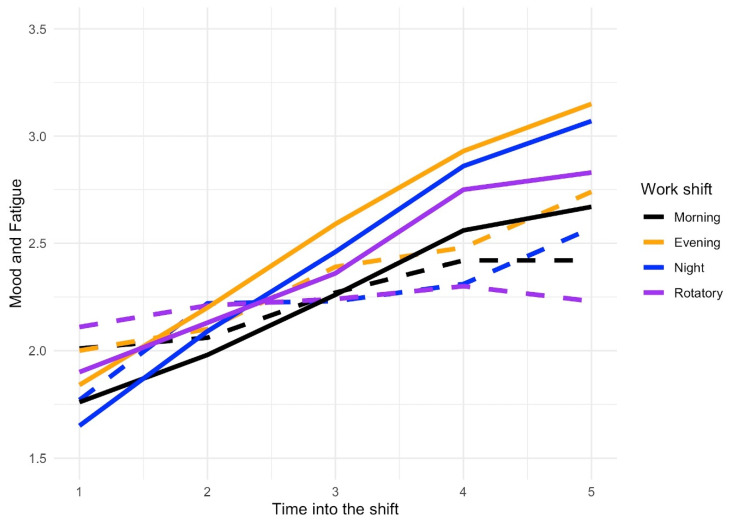
Evolution of mood and fatigue during the shift: Continuous lines represent mood and dashed lines represent fatigue. The higher values reflect negative mood and high fatigue. The range 1–5 of the dependent variables is limited to 1.5–3.5 for better visualization.

**Figure 3 ijerph-17-07277-f003:**
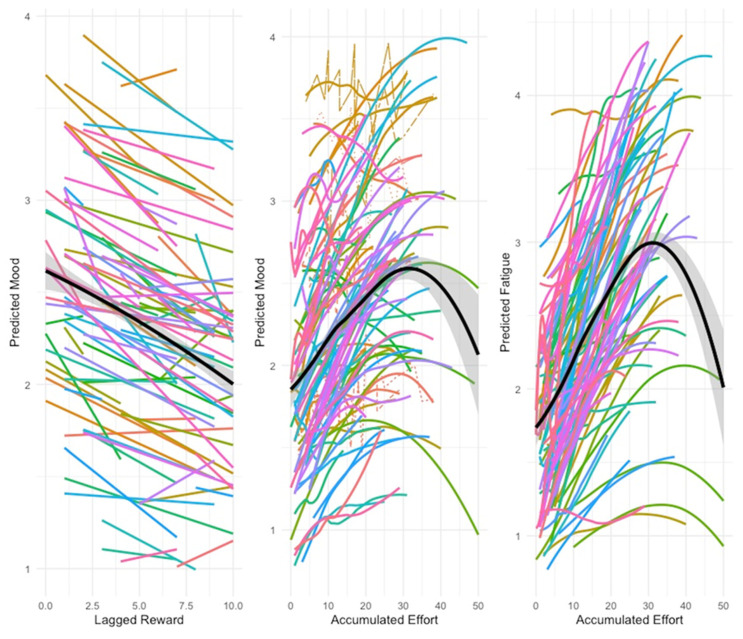
Spaghetti plots of fitted models with quadratic trend variables. Every line represents an individual’s trajectory, and the bold black lines represent the smooth means, with the 95% confidence intervals in grey.

**Table 1 ijerph-17-07277-t001:** Fixed-effects estimates, variance estimates: standard error and the fit indices for the fitted models of the predictors of lagged and accumulated effects on mood and fatigue.

Stress				
	Mood		Fatigue	
**Parameter**	*Lagged*	*Accumulated*	*Lagged*	*Accumulated*
Fixed effects				
Intercept	1.53(9.67 × 10^−2^) ***	2.44(0.10) ***	1.35(1.11) ***	1.95(9.74 × 10^−2^) ***
**Level 1 (Moment)**				
Time Demand Control Effort Reward	Time: 8.63 × 10^−2^(2.12 × 10^−2^) *** Effort: 9.40 × 10^−2^(8.57 × 10^−3^) *** Reward lag1 (quadratic): 2.68(1.00) **	Time: −0.07(0.02) * Effort (quadratic): −3.92(0.95) ***	Time: 2.82 × 10^−1^(2.48 × 10^−2^) *** Effort lag1: 2.15 × 10^−2^(8.28 × 10^−3^) ** Effort: 4.95e−02(8.19 × 10^−3^) *** Reward lag1: −2.71 × 10^−2^(8.95 × 10^−3^) **	Time: 2.41 × 10^−1^(3.61 × 10^−2^) *** Effort (quadratic): −2.35(8.54 × 10^−1^) ** Reward: −1.79 × 10^−2^(3.78 × 10^−3^) ***
Random effects				
**Level 1 (Moment)**				
Intercept SD	0.59	0.73	0.61	0.72
Time SD	0.11	0.15	0.18	0.20
ICC	0.43	0.43	0.51	0.50
AIC	−	4641	3329	4169
BIC	−	4686	3376	4219
**Coping**				
Fixed effects				
Intercept	1.80(9.29 × 10^−2^) ***	1.95(8.51 × 10^−2^) ***	1.46(9.00 × 10^−2^) ***	1.54(0.08) ***
**Level 1 (Moment)**				
Time Problem-focused Emotion-focused Seeking support Refusal	Time: 9.32 × 10^−2^(2.16 × 10^−2^) *** Problem: 3.46 × 10^−1^(5.75 × 10^−2^) *** Emotion: 2.24 × 10^−1^(5.47 × 10^−2^) ***	Time: 6.89 × 10^−2^(2.39 × 10^−2^) ** Emotion: 7.57 × 10^−2^(2.60 × 10^−2^) ** Refusal: 2.86 × 10^−1^(7.04 × 10^−2^) ***	Time: 2.80 × 10^−1^(2.50 × 10^−2^) *** Problem lag1: 9.90 × 10^−2^(5.05 × 10^−2^) * Emotion: 1.72 × 10^−1^(4.97 × 10^−2^ ) *** Refusal: 5.32 × 10^−1^(1.287 × 10^−1^) ***	Time: 0.24(0.02) *** Problem: 0.04(0.02) * Emotion: 0.08(0.02) *** Support: −0.11(0.03) ** Refusal: 0.27(0.06) ***
Random effects				
**Level 1 (Moment)**				
Intercept SD	0.59	0.73	0.63	0.72
Time SD	0.11	0.17	0.18	0.21
ICC	0.41	0.42	0.52	0.52
AIC	3572	4693	3352	4208
BIC	3615	4738	3400	4263
Task				
Fixed effects				
Intercept	1.96(9.17 × 10^−2^) ***	1.94(0.08) ***	1.48(0.08) ***	1.51(8.15e−02) ***
**Level 1 (Moment)**				
Time Direct Care Indirect Care Medication Documentation Communication Social/resting	Time: 8.54 × 10^−2^(2.16 × 10^−2^) *** Direct Care lag1: 1.30 × 10^−1^(4.53 × 10^−2^) **	Time: 0.11(0.02) *** Documentation:−0.09(0.03) **	Time: 0.29(0.02) *** Direct Care: 0.20(0.04) *** Documentation lag1: −0.12(0.05) * Communication lag1: −0.22(0.08) **	Time: 2.15e−01(2.54 × 10^−2^) *** Direct Care: 1.76 × 10^−1^(2.34 × 10^−2^) *** Medication: 1.25 × 10^−1^(2.92 × 10^−2^) ***
Random effects				
**Level 1 (Moment)**				
Intercept SD	0.62	0.73	0.64	0.71
Time SD	0.11	0.16	0.18	0.20
ICC	0.42	0.42	0.52	0.52
AIC	3607	−	3349	4182
BIC	3644	−	3397	4227

Note: Standard errors are in parenthesis. ICC = Intraclass Correlation Coefficient; AIC = Akaike Information Criterion; BIC = Bayesian Information Criterion; * *p* < 0.05; ** *p* < 0.01; *** *p* < 0.001; n.s. = non-significant.

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
