# Peer review of "How the Lagged and Accumulated Effects of Stress, Coping, and Tasks Affect Mood and Fatigue during Nurses’ Shifts"

_ijerph, 2020, doi:10.3390/ijerph17197277_

Round 1

Reviewer 1 Report

I congratulate the authors for the article presented. It is a powerful object of study with contributions in the area of ​​public health and nursing.

I point out some observations that need more attention in the sense of corrections.

About the abstract, be more objective. Detail the method better and be specific in the conclusion. Point out the results objectively.

In the introduction, be specific, present the object of study, justify the importance of the study and finish with the objectives. Please avoid repetition of ideas (line 51-86). It is necessary to join the highlighted points of line 108 to 125.

Start the method by describing the type of study. I suggest following the STROBE protocol.
Was the study a sample? Which calculation is used? Was it a sample of convenience? This sample seems fragile to me.
I suggest finalizing the method with ethical aspects.
Explain better what is the concept of Momentary Ecological Assessment.

The results are adequate and speak to the method.

The discussion is fragile, does not support the results. I miss a comparison with other studies. The limitations were mentioned, however, what were the elements used to overcome the limitations: describe!
I miss a deep discussion about routine social support.

The conclusion meets the objectives of the study and meets the standards of the journal.

References need adjustment. They do not meet the magazine's standards.

Author Response

Reviewer #1: I congratulate the authors for the article presented. It is a powerful object of study with contributions in the area of ​​public health and nursing.

I point out some observations that need more attention in the sense of corrections.

Thank you very much, your comments are accurate and useful to improve the writing.

About the abstract, be more objective. Detail the method better and be specific in the conclusion. Point out the results objectively.

Thank you for your recommendation. Yes, the abstract seemed a bit confusing and it has been restructured, incorporating more details about methods, ordering the results and presenting more information in the conclusions.

In the introduction, be specific, present the object of study, justify the importance of the study and finish with the objectives. Please avoid repetition of ideas (line 51-86). It is necessary to join the highlighted points of line 108 to 125.

The introduction has been changed in order to be more specific and related to the objectives, including a final paragraph of transition to objectives. Thank you very much for your suggestion, which improves reading.

Start the method by describing the type of study. I suggest following the STROBE protocol.

Some changes have been made to offer a more closed format to STROBE protocol, but the information about variables is included in the statistical methods for clarity due to the special characteristics of the multilevel modelling structure.

Was the study a sample? Which calculation is used? Was it a sample of convenience? This sample seems fragile to me.

Nurses who were offered to participate were randomly selected, but stratified by ward.Of course, the hospitals that participated were “of convenience” in the sense that they were the hospitals of the Universities from de groups engaged in this research project.

The sample of the study was calculated trying to reach the 80% of the whole ward nurses in the hospitals.

Representative calculations in samples for multilevel modelling are a complicate matter. As stated by Meinck and Vandenplas (2012), the literature review shows that knowledge about how best to determine sample sizes when using multilevel modeling for data analysis is still developing. It is difficult to find any definite or generalized solutions regarding this matter. It is important to have in mind that we have sample sizes at different levels. 

Simulation calculations or specific complex software have been developed to solve this problem, but other authors recommend  some rules of thumb. Hox (2010, p. 235) proposes the following rule of thumb, used in the present study:

  • 30/30: 30 groups and 30 observations per group. If the interest is focused on the fixed parameters
  • 50/20: 50 groups and 20 observations per group. If there is strong interest in cross-level interactions.
  • 100/10: 100 groups with at least 10 observations per group. If there is a strong interest in the random part.

Our design set up two levels: Level 1 (within person) includes moments, nested participants (level 2, within person). Level 1 included an average of 5 records and there were recorded 5 shifts; so there were 25 observations per participant. Lastly, our study was interested in cross-level interactions and random effects. Taking all this considerations as a whole, we tried to reach to at least 100 nurses (we selected 113 nurses and obtained 96 participants, finally), this was a 100/25 rule.

I suggest finalizing the method with ethical aspects.

The Ethics section is now at the end of the Methods section.

Explain better what is the concept of Momentary Ecological Assessment.

A paragraph at the beginning of the Measures section has been added in order to introduce and explain the new concept of Ecological Momentary Assessment (EMA).

The results are adequate and speak to the method.

Thank you.

The discussion is fragile, does not support the results. I miss a comparison with other studies. The limitations were mentioned, however, what were the elements used to overcome the limitations: describe!

We have reviewed the discussion trying to strengthen it. We have added references to other studies. But it should be considered that the fact of making this momentary evaluation recording the specific task that is being done does not allow much comparison with the large amount of literature that we have in which data are collected through general questionnaires that are not so specific . For example, the frequency of use of moment-to-moment coping strategies cannot be compared with the frequencies of use of coping strategies that are recorded by asking about the general frequency in the last 15 days. We hope that the modifications introduced are satisfactory.

The limitation for our study is that the results are not generalizable to other type of nurses, but the majority of registered nurses working in the hospital are ward nurses, so this study covers quite well their professional profile. A brief commentary is now included in the text.

I miss a deep discussion about routine social support.

We have now added a paragraph in the Discussion, with recommendations from the American Organization for Nursing Leadership, as well as a theoretical reference on the integration of social support in daily practice.

The conclusion meets the objectives of the study and meets the standards of the journal.

Thank you.

References need adjustment. They do not meet the magazine's standards.

The ‘Instructions for authors’ section in the journal is confusing because it says first that ‘Your references may be in any style, provided that you use the consistent formatting throughout’ (https://www.mdpi.com/journal/ijerph/instructions), and later the same text points to a specific format style.

We have changed finally the references into the specific format style of the journal. Thank you for your recommendation.

Hox, J.J. (2010). Multilevel analysis: Techniques and applications. Routledge, New York, NY.

Meinck, S. & Vandenplas, C. (2012). Sample Size Requirements in HLM: An Empirical Study. Large-Scale Assessments in Education Special Issue 1.

Reviewer 2 Report

This article report on a well developed study on the effects of efforts, rewards, coping and tasks on mood and fatigue in nurses. The design of the study is robust.

There are only a few minor points to comment on:

Introduction. The background is well described and there is an updated exposition of the state-of-the-art of the main topic (fatigue, mood and nursing tasks).

The aims and hypothesis are clearly exposed.

Methods

- pg 4 line 141. It is not clear “96 nurses with a response rate of 85.47%” ( To which refers this response rate?

- line 150. Data collection procedure. It seems that a smartphone was used for data collection, but it is not totally clear. Each nurse who participated in the study was provided with one of these smartphones? How many time / days / shifts did each nurse carry the smartphone? Please, explain a little more, so readers can understand it well.

  • 4.2.2 Questionnaires. Lines 197. When were the questionnaire administered to the nurses?

- Discussion. The main findings of the study are well explained and clearly organized by aims.

References:

  • Duplicate reference: Johnston et al ( 2019ª ) , (2019b), line 512 and 516. Please, check.

Author Response

Reviewer #2: This article report on a well developed study on the effects of efforts, rewards, coping and tasks on mood and fatigue in nurses. The design of the study is robust.

There are only a few minor points to comment on:

Introduction. The background is well described and there is an updated exposition of the state-of-the-art of the main topic (fatigue, mood and nursing tasks).

The aims and hypothesis are clearly exposed.

Thank you.

Methods

- pg 4 line 141. It is not clear “96 nurses with a response rate of 85.47%” ( To which refers this response rate?

Yes, the final sample was 96 nurses, from possible 113. We have added a small clarification between brackets at the end of the paragraph in the text (’96 of possible 113’). Thank you for your comment.

- line 150. Data collection procedure. It seems that a smartphone was used for data collection, but it is not totally clear. Each nurse who participated in the study was provided with one of these smartphones? How many time / days / shifts did each nurse carry the smartphone? Please, explain a little more, so readers can understand it well.

Each nurse was provided with a smartphone and it was programmed with her/him for the next 5 work shifts (including weekend if working). She/he carried the device during the time between work shifts too. The smartphone was only used to collect the data. It had no phone line, nor could it be used for any other function. We had several of them, we gave them to the nurses and when the days of registration were over, they returned it and we gave it to another nurse. This was an especially important procedure because nurses in Spain tend to change work shifts with other colleagues quite frequently. We have added a small comment after the reference to the smartphone device indicating that all the measures were taken with it and the time they carried it.

  • 4.2.2 Questionnaires. Lines 197. When were the questionnaire administered to the nurses?

The ad-hoc questionnaire was administered just before the smartphone was programmed, in a room prepared for it.

- Discussion. The main findings of the study are well explained and clearly organized by aims.

Thank you.

References:

  • Duplicate reference: Johnston et al ( 2019ª ) , (2019b), line 512 and 516. Please, check.

Thank you. The duplication has been removed.

Round 2

Reviewer 1 Report

I congratulate the authors for the important scientific production.